# Clinical Characteristics of Snakebite Envenomings in Taiwan

**DOI:** 10.3390/toxins17010014

**Published:** 2024-12-30

**Authors:** Yan-Chiao Mao, Po-Yu Liu, Kuo-Lung Lai, Yi Luo, Kuang-Ting Chen, Chih-Sheng Lai

**Affiliations:** 1Department of Medical Toxicology, Taichung Veterans General Hospital, 1650 Taiwan Boulevard Sect. 4, Taichung 407204, Taiwan; nilnil.nilnil@vghtc.gov.tw; 2PhD Program in Medical Biotechnology, College of Medical Science and Technology, Taipei Medical University, No.250, Wuxing St., Xinyi Dist., Taipei 110301, Taiwan; 3School of Medicine, National Defense Medical Center, No.161, Sec. 6, Minquan E. Rd., Neihu Dist., Taipei 114201, Taiwan; 4Department of Post-Baccalaureate Medicine, College of Medicine, National Chung Hsing University, 145 Xingda Rd., South Dist., Taichung 402202, Taiwan; 5Department of Internal Medicine, Division of Infecious Diseases, Taichung Veterans General Hospital, 1650 Taiwan Boulevard Sect. 4, Taichung 407204, Taiwan; idfellow@gmail.com; 6Rong Hsing Research Center for Translational Medicine, National Chung Hsing University, 145 Xingda Rd., South Dist., Taichung 402202, Taiwan; 7Department of Internal Medicine, Division of Allergy, Immunology and Rheumatology, Taichung Veterans General Hospital, 1650 Taiwan Boulevard Sect. 4, Taichung 407204, Taiwan; kllaichiayi@yahoo.com.tw; 8Liuzhou Integrated Chinese and Western Medicine Snake Injury Treatment Center, Liuzhou Traditional Chinese Medicine Hospital, Liuzhou 545001, China; 13877288998@163.com; 9Department of Chinese Medicine, Chang Bing Show Chwan Memorial Hospital, No. 6, Lugong Rd., Lukang Township, Changhua County 505029, Taiwan; u102022410@cmu.edu.tw; 10Graduate Institute of Biomedical Sciences, China Medical University, No. 100, Section 1, Jingmao Road, Beitun District, Taichung 406040, Taiwan; 11Department of Surgery, Division of Plastic and Reconstructive Surgery, Taichung Veterans General Hospital, 1650 Taiwan Boulevard Sect. 4, Taichung 407204, Taiwan

**Keywords:** venomous snakes, venom, clinical manifestations, local signs, systemic signs, complications of envenoming

## Abstract

Snakebite envenomings continue to represent a major public health concern in Taiwan because of the presence of various venomous snakes whose habitats intersect with human activities. This review provides a comprehensive analysis of the clinical characteristics, complications, and management strategies associated with snakebite envenomings in Taiwan. Taiwan is inhabited by six principal venomous snakes: *Trimeresurus stejnegeri stejnegeri*, *Protobothrops mucrosquamatus*, *Deinagkistrodon acutus*, *Daboia siamensis*, *Naja atra*, and *Bungarus multicinctus*, each presenting distinct clinical challenges. The clinical manifestations vary from local symptoms such as pain, swelling, and necrosis to systemic complications including neurotoxicity, coagulopathy, and organ failure, depending on the species. Notable complications arising from these snakebite envenomings include necrotizing soft tissue infection, compartment syndrome, respiratory failure, and acute kidney injury, often necessitating intensive medical interventions. This review highlights the critical importance of early diagnosis, the prompt administration of antivenoms, and multidisciplinary care to improve patient outcomes and reduce healthcare costs. Future research is encouraged to enhance treatment efficacy, improve public awareness, and develop targeted prevention strategies. By identifying gaps in current knowledge and practice, this work contributes to the global literature on envenoming management and serves as a foundation for advancing clinical protocols and reducing snakebite-related morbidity and mortality in Taiwan.

## 1. Introduction

Taiwan is home to six venomous snake species of medical importance [1,2]. *Trimeresurus stejnegeri stejnegeri* (bamboo pit viper), *Protobothrops mucrosquamatus* (Taiwan habu), and *Deinagkistrodon acutus* (hundred pacer) are members of the Viperidae family, specifically the subfamily Crotalinae [1,2]. Additionally, the Viperidae family includes *Daboia siamensis* (Russell’s viper), which belongs to the subfamily Viperinae. The Elapidae family comprises two species: *Naja atra* (Chinese cobra) and *Bungarus multicinctus* (many-banded krait) [1,2]. These species are distributed across Taiwan with various habitat preferences. *T*. *s*. *stejnegeri*, *P*. *mucrosquamatus*, and *B*. *multicinctus* are found throughout the island, whereas *N*. *atra* is predominantly located on the Central Plain. By contrast, *D*. *acutus* and *D*. *siamensis* have a more dispersed distribution, with *D*. *siamensis* mainly inhabiting the southern and eastern regions of the country [1,2]. In addition to the six major venomous species, recent reports have indicated that two additional rare crotaline species, *Ovophis makazayazaya* and *T*. *gracilis*, may pose significant envenoming risks to humans [3,4], each exhibiting distinct physical characteristics (Figure 1).

Snakebite envenomings in Taiwan constitute a major public health concern, primarily due to the intersection of snake habitats with human activities such as farming and forestry [5]. If patients without a physical snake specimen are assessed for antivenom administration, clinical symptoms and signs of envenomation should serve as the primary determinants for its use. This decision typically relies on a combination of clinical manifestations, the characteristics of the envenoming site, patient-reported information about the snake’s appearance, and local epidemiological data on venomous snakes. Additionally, laboratory findings, such as coagulopathy markers, may guide the decision. These updates highlight the diagnostic strategies employed in such scenarios, ensuring the provision of appropriate and effective treatment. The clinical impact of snakebite envenomings can vary considerably depending on the species involved. Envenomings by elapids such as *N*. *atra* and *B*. *multicinctus* lead to cytotoxic and neurotoxic symptoms, such as wound necrosis and paralysis, respectively, which can be fatal if not promptly managed through surgical intervention or intensive care [5,6,7]. Conversely, envenomings by crotalines, including *T*. *s*. *stejnegeri* and *P*. *mucrosquamatus*, primarily cause local tissue injury characterized by swelling, pain, ecchymosis, and necrosis. However, these envenomings can occasionally lead to systemic effects such as coagulopathy, rhabdomyolysis, and acute kidney injury [6].

The complications that arise following venomous snakebites are varied and depend on factors such as the snake species, the quantity of venom injected, and the promptness and appropriateness of medical intervention [7]. Common acute complications include necrotizing soft tissue infection, compartment syndrome, and systemic effects such as neurotoxicity, coagulopathy, and acute kidney injury. Long-term complications include chronic pain, loss of limb function, and psychological disorders.

The incidence of snakebites in Taiwan was estimated at 4.2 cases per 100,000 individuals [7]. The healthcare costs associated with snakebite envenomings in Taiwan are substantial, exerting a considerable strain on the National Health Insurance system [8]. These costs are driven by several factors, including the need for antivenom, hospitalization, intensive care, and surgical interventions (e.g., fasciotomies, debridement, and amputations) [9,10] and long-term rehabilitation. A study examining healthcare costs related to snakebite cases in Taiwan revealed substantial variations in the mean cost per patient depending on the severity of the envenoming and the extent of the medical interventions required [8]. For example, patients requiring intensive care due to respiratory failure or multiple surgical interventions to address wound necrosis and infection can incur high healthcare costs. Additionally, the financial burden is exacerbated by indirect costs, such as loss of productivity and long-term disability.

Snakebite envenomings remain a significant yet underrecognized public health issue in Taiwan, with limited comprehensive reviews. Existing studies primarily focus on isolated case reports or specific aspects of envenoming, leaving a gap in synthesized knowledge that could guide clinical practice and public health strategies. The primary objective of this review is to provide a detailed examination of the clinical characteristics of different snakebite envenomings in Taiwan, with a focus on understanding their presentation and management. By consolidating and analyzing available data, this review aims to bridge the knowledge gap, offering practical insights for healthcare providers and policymakers to improve patient outcomes and optimize resource allocation. Additionally, it not only serves as a comprehensive reference for clinicians managing snakebite cases but also informs public health initiatives aimed at reducing morbidity and mortality associated with envenomings. The findings contribute to the global literature by providing a region-specific perspective on snakebite management, with potential implications for other areas facing similar challenges. This review addresses the key challenges in diagnosing and managing snakebite envenomings and proposes areas for future research to enhance patient care and clinical outcomes in this critical domain of tropical medicine. The inclusion criteria for the reviewed literature focused on studies published in peer-reviewed journals, case reports, and regional data relevant to snakebite envenomings in Taiwan. The literature was selected based on its clinical relevance, contribution to understanding envenoming presentations, and applicability to management practices.

## 2. Clinical Characteristics of *Trimeresurus stejnegeri stejnegeri* Envenoming

*T*. *s*. *stejnegeri*, commonly known as Stejneger’s bamboo pit viper, is one of the venomous snakes most frequently encountered in Taiwan and accounts for nearly half of all reported snakebite envenomings annually [11,12]. This species predominantly has localized envenoming effects, although systemic complications can occur in certain cases [12].

The clinical presentation of *T*. *s*. *stejnegeri* envenoming is characterized by a rapid onset of local symptoms at the bite site, with swelling (100%), pain (100%), and ecchymosis (22%) (Figure 2) being the most common manifestations [6]. In some cases, bullae or blisters can form, accompanied by the development of lymphangitis and lymphadenitis [13]. These symptoms typically develop within hours of the bite and can spread beyond the affected limb, extending into the trunk and leading to considerable morbidity [13,14]. The local effects are exacerbated by the venom, which contains a complex mixture of metalloproteinases, phospholipases, and serine proteases that together contribute to and perpetuate tissue injuries [6,13].

Although primarily associated with local tissue injuries, *T*. *s*. *stejnegeri* envenoming can occasionally result in systemic coagulopathy, a less common but potentially life-threatening complication [13,14,15]. This condition is characterized by hypofibrinogenemia, a long prothrombin time (PT), and elevated levels of fibrin degradation products (FDPs) and D-dimer, all indicative of consumptive coagulopathy. These coagulation abnormalities arise from the fibrinogenolytic activity of the snake’s venom, which causes defibrinogenation and disrupts hemostasis [14,15]. Clinically, this may manifest as spontaneous bleeding or, in severe cases, hemorrhagic shock [14,15].

*T*. *s*. *stejnegeri* envenoming requires the prompt administration of bivalent antivenom, which is produced by the Taiwan Centers for Diseases Control and is effective against both *T*. *s*. *stejnegeri* and *P*. *mucrosquamatus*. The initial treatment typically involves the administration of one to two vials of antivenom, with additional doses administered as necessary, depending on the severity of the symptoms and laboratory findings [2]. In cases where coagulopathy is observed, fresh frozen plasma (FFP) and cryoprecipitate may be administered to replenish coagulation factors and control active bleeding. The close monitoring of the patient’s coagulation profile—including PT, activated partial thromboplastin time (aPTT), fibrinogen, FDPs, D-dimer levels, and platelet count—is crucial for assessing the effectiveness of the treatment and guiding further therapeutic decisions [14,15].

Based on clinical experience, patients envenomed by *T*. *s*. *stejnegeri* generally have favorable outcomes when treated promptly and appropriately with antivenom and supportive care [8]. However, delays in seeking medical attention or inadequate initial treatment may result in complications such as tissue necrosis, secondary infection, and prolonged hospitalization.

## 3. Clinical Characteristics of *Protobothrops mucrosquamatus* Envenoming

*P*. *mucrosquamatus*, commonly referred to as the Taiwan habu or brown spotted pit viper, is a medically significant venomous snake in Taiwan [2]. Envenoming by this species predominantly results in local tissue injuries rather than systemic complications. Unlike some other venomous snakes, such as *T*. *s*. *stejnegeri*, *P*. *mucrosquamatus* rarely induces serious coagulopathy, making its effects largely localized [10,16].

Patients envenomed by *P*. *mucrosquamatus* typically present with rapid-onset tissue swelling and tenderness (Figure 3) that can progress to involve the entire limb if not promptly managed [16]. This swelling is primarily due to the venom’s myotoxic and cytotoxic components, which compromise the vascular endothelium, leading to increased permeability and subsequent tissue swelling. In some cases, the venom may also induce myositis, a rare but severe complication characterized by muscle inflammation and necrosis [17]. Diagnostic imaging, such as ultrasonography [16], is frequently employed to assess the extent of muscle involvement and detect early signs of compartment syndrome, a potential complication that requires surgical intervention.

The management of *P*. *mucrosquamatus* envenoming involves the administration of bivalent antivenom [10]. The recommended initial dose is typically 2–4 vials, depending on the severity of local symptoms. Additional doses may be required if the patient’s condition does not improve or if swelling or pain continues to progress. In addition to antivenom therapy, supportive care measures—including analgesia, adequate hydration, and, in cases of tissue necrosis, surgical debridement—may be required to effectively manage local tissue damage [16].

In a prospective observational study conducted between 2017 and 2022, 29 patients bitten by *P*. *mucrosquamatus* were monitored using point-of-care ultrasound to assess the rate of the proximal progression (RPP) of the swelling [16]. The study revealed that patients with a lower RPP had more favorable outcomes and required fewer vials of antivenom compared with those with a higher RPP. These findings suggest that careful monitoring of the RPP can guide antivenom administration decisions, potentially reducing the risk of overtreatment and associated complications, such as serum sickness, while also reducing healthcare costs.

Overall, although *P*. *mucrosquamatus* envenoming rarely results in systemic complications, the local tissue injury can be severe and necessitate prompt and appropriate medical intervention. Imaging modalities such as ultrasonography have proven instrumental in assessing the extent of envenoming and guiding treatment strategies [16]. With appropriate management, including the prudent administration of antivenom and comprehensive supportive care, most patients achieve favorable outcomes without significant long-term complications, as observed in clinical practice [8].

## 4. Clinical Characteristics of *Deinagkistrodon acutus* Envenoming

Envenoming by *D*. *acutus* is rare but highly dangerous due to this species’ potent hemotoxic venom, which can lead to coagulopathy, thrombocytopenia, and extensive tissue necrosis. Clinical manifestations typically begin within 2–3 h of the bite and include severe pain, swelling, and the development of hemorrhagic bullae (Figure 4) around the bite site [7,17,18,19]. The systemic symptoms are often coagulopathy, characterized by markedly increased PT and aPTT, along with severe thrombocytopenia. Laboratory findings may reveal an international normalized ratio of PT that is >9 and a platelet count of <50,000/mm³, indicating the impairment of blood clotting mechanisms [20]. In severe cases, patients may present with incoagulable blood, spontaneous bleeding, and even hemorrhagic shock if not promptly treated [21].

Regarding management, *D*. *acutus* envenoming requires the immediate administration of specific antivenom to neutralize the venom and mitigate its systemic effects. The recommended initial dose is 2–4 vials of monovalent antivenom for *D*. *acutus*, with additional doses administered every 6–8 h if coagulopathy or thrombocytopenia persists [21]. In cases of severe coagulopathy and thrombocytopenia, adjunctive therapies such as FFP and platelet transfusion may be considered, particularly when considerable bleeding occurs, or the platelet level is critically low.

Delayed or inadequate administration of antivenom heightens the risk of severe complications, such as compartment syndrome, necrosis requiring surgical intervention, and, in extreme cases, multiorgan failure. Even with the prompt administration of antivenom and supportive care, some patients may require a long period of hospitalization and multiple surgical procedures due to wound necrosis and secondary infections [17,19,20]. Effective management also involves vigilant monitoring for potential complications, such as systemic bleeding and acute kidney injury, which can arise from the profound systemic effects of the venom [20,21].

## 5. Clinical Characteristics of *Daboia siamensis* Envenoming

Envenoming by *D*. *siamensis* is particularly concerning due to the potent hemotoxic effects of this species’ venom, which can induce coagulopathies. Patients bitten by *D*. *siamensis* often present with immediate and severe local symptoms, such as swelling, pain, and ecchymosis (Figure 5) at the bite site [22,23,24]. The venom of Russell’s viper contains a complex mixture of toxins, including factors that activate prothrombin and damage endothelial cells, leading to increased vascular permeability and coagulopathy. This leads to a cascade of systemic effects, notably consumptive coagulopathy, hypofibrinogenemia, and modest thrombocytopenia [20]. These coagulopathies can manifest as spontaneous bleeding from mucosal sites, hematuria, gastrointestinal bleeding, and intracranial hemorrhage, all of which require urgent medical intervention.

Laboratory findings in patients envenomed by *D*. *siamensis* typically reveal an increased PT and aPTT, a decreased platelet count, a decreased fibrinogen level, and increased D-dimer levels [20]. These findings indicate the ongoing activation of the coagulation cascade and consumption of coagulation factors. Extremely high levels of D-dimer and a marked depletion of fibrinogen are particularly indicative of *D*. *siamensis* envenoming, distinguishing it from other snakebite envenomings, such as those caused by *D*. *acutus*, which may present as moderate thrombocytopenia but not the same degree of fibrinogen consumption [20].

Regarding the management of *D*. *siamensis* envenoming in Taiwan, the emphasis is on the rapid administration of specific antivenom to neutralize the circulating venom and prevent the further progression of coagulopathy [24]. Due to the severity of the venom’s hemotoxic effects, the initiation of antivenom therapy must be prompt. In addition to antivenom therapy, supportive treatments such as transfusions of blood products (e.g., platelets, FFP, and cryoprecipitate) may be necessary to manage active bleeding and correct coagulation deficits [23,25].

## 6. Clinical Characteristics of *Naja atra* Envenoming

*N*. *atra*, commonly known as the Chinese cobra, is a medically important venomous snake in Taiwan and is responsible for numerous human envenomings annually [11]. The clinical manifestations of *N*. *atra* envenoming are severe local symptoms due to the cytotoxic components of the species’ venom. Patients typically present with localized pain, swelling, and notable tissue necrosis, which occur in approximately 65.6% of cases (Figure 6) [26,27,28]. Although the venom also contains neurotoxins, it does not typically produce considerable neurotoxicity in humans [29].

Upon *N. atra* envenoming, the local effects are rapid, often occurring within 30 min to a few hours. The bite site typically exhibits erythema, swelling, and pain, which can progress to wet necrosis and secondary bacterial infection if not promptly managed [26]. The cytotoxic nature of *N*. *atra* venom results in the destruction of cellular membranes and substantial tissue necrosis [29]. In severe cases, surgical intervention, such as debridement or even the amputation of digits or toes, may be required to manage necrotic tissue and prevent further damage [10,30,31,32].

The management of *N*. *atra* envenoming involves the administration of antivenom, which is effective in neutralizing the circulating venom [26]. However, concerns remain regarding its efficacy against tissue-bound cardiotoxins [33]. Surgical intervention is often necessary as an adjunct therapy to more effectively manage necrotic wounds and secondary bacterial infections [7,8,30]. Broad-spectrum antibiotics are commonly prescribed to cover potential pathogens, including Gram-negative bacteria such as *Shewanella algae* and *Morganella morganii*, which have been implicated in secondary infections following *N*. *atra* bites [34,35,36,37]. Regarding the challenges associated with multidrug-resistant (MDR) pathogens in snakebite wounds, it is crucial to emphasize the need for a tailored antibiotic therapy guided by culture and sensitivity results whenever feasible. Furthermore, promoting antimicrobial stewardship is essential to mitigate the risk of resistance development while ensuring effective infection management.

*N*. *atra* envenoming poses a crucial clinical challenge because the initial wound symptoms can resemble those caused by other common crotaline snakebites. Early recognition, coupled with appropriate surgical and supportive care, is crucial for effective management and improved clinical outcomes. Continued research into the epidemiology, venom properties, and effective treatments for *N*. *atra* envenoming is crucial for reducing the morbidity and mortality associated with these bites in Taiwan.

## 7. Clinical Characteristics of *Bungarus multicinctus* Envenoming

*B*. *multicinctus* is renowned for its potent neurotoxic venom, which primarily affects the nervous system without causing major local tissue damage. Patients envenomed by *B*. *multicinctus* typically present with minimal local symptoms at the bite site, such as slight swelling (Figure 7) and pain [19,38,39]. However, the venom’s neurotoxic effects lead to progressive systemic symptoms that usually develop within a few hours of the bite [38,39,40]. Early signs of envenoming include ptosis (drooping of the eyelids), ophthalmoplegia (paralysis or weakness of the eye muscles), dysarthria (difficulty speaking), and generalized muscle weakness, which often manifests as descending paralysis. In severe cases, the condition can progress to respiratory muscle paralysis, necessitating mechanical ventilation to prevent respiratory failure [7,8,38,39,40].

In *B. multicinctus* envenoming, the bite may not appear immediately hazardous, meaning that initial medical attention is often delayed. The underestimation of the severity of krait bites can result in delayed hospital presentation and a long hospital stay [18,40]. The neurotoxic effects of the venom can escalate rapidly, making early recognition and the prompt administration of specific antivenom crucial for achieving the optimal patient outcomes. Antivenom therapy, which effectively neutralizes the neurotoxins, is the primary treatment and is most beneficial when administered within the first few hours after the bite [2].

Even if they receive antivenom treatment, patients envenomed by *B. multicinctus* may have prolonged neurological symptoms, necessitating extended supportive care, including respiratory support and intensive monitoring. Patients should be closely observed for at least 8 h to detect any delayed onset of neurotoxic symptoms. Early medical intervention markedly improves the prognosis for *B. multicinctus* envenoming, but the risk of neurological complications, such as polyneuropathy, remains high [38,39,40].

## 8. Clinical Characteristics of Uncommon Crotaline Envenomings from *Ovophis makazayazaya* and *Trimeresurus gracilis* Bites

*O*. *makazayazaya*, commonly known as the Taiwan mountain pit viper, is a species indigenous to the mountainous regions of Taiwan. Although envenomings by this species are rare, they can lead to considerable clinical manifestations. The primary symptoms are typically severe localized pain, progressive swelling, and, in some cases, hemorrhagic bullae at the bite site [3]. For instance, in one reported case, swelling progressed from the foot to the thigh during the 1–2 days following the bite [3]. Laboratory findings in these cases typically reveal an elevated white blood cell count and abnormal coagulation profile, such as a long PT and elevated D-dimer level, indicating potential coagulopathy [4]. Antivenom specific to *O*. *makazayazaya* is not available; therefore, treatment usually involves the administration of bivalent antivenoms designed for other pit viper species, such as *T*. *s*. *stejnegeri* and *P*. *mucrosquamatus*, which may offer cross reactivity and help mitigate the effects of the venom [3].

*T*. *gracilis*, commonly known as the Kikuchi habu, is another pit viper species native to Taiwan, inhabiting altitudes above 2000 m. The clinical manifestations of envenoming by this species include localized pain, swelling, and ecchymosis along with the formation of hemorrhagic bullae [4]. In some cases, compartment syndrome may develop, necessitating surgical intervention such as fasciotomy. Coagulopathy, characterized by an elevated D-dimer level and a long PT, has been observed in patients; it potentially leads to delayed wound healing and, in severe cases, permanent deformities in the affected limb [5]. Even if large volumes of bivalent antivenom are administered, the cross-neutralizing efficacy against *T*. *gracilis* venom is insufficient, and additional medical and surgical interventions are often required [4].

Overall, envenoming by *O. makazayazaya* and *T. gracilis* presents considerable clinical challenges due to the absence of species–specific antivenoms. Clinicians must thus rely on cross-reactive antivenoms and comprehensive medical management to prevent complications such as necrosis, coagulopathy, and limb deformities.

The stark differences between Taiwan and other regions (southern Burkina Faso and West Africa) underscore the critical role of robust healthcare systems, national insurance coverage, and accessibility to effective antivenom in mitigating the financial and health impacts of snakebite envenomation (SBE) [41,42]. While Taiwan’s economic burden per case is higher, it reflects a more advanced healthcare system capable of delivering comprehensive care, including the management of complications [8,41,42].

In conclusion, we reviewed the diagnostic complexities arising from overlapping symptoms caused by six principal venomous snake species, as well as rare crotaline species such as *Ovophis makazayazaya* and *Trimeresurus gracilis*. In Taiwan, the mortality rate associated with snakebite envenoming is relatively low due to the accessibility of medical care and the availability of effective antivenoms. Based on the epidemiological data, the mortality rate has been reported to range between 0.1% and 0.5% for hospitalized cases [5]. The species most frequently associated with severe outcomes, including fatalities, is *Bungarus multicinctus* (many-banded krait) [7,40]. Its potent neurotoxic venom can lead to respiratory failure if not treated promptly. Other species, such as *Naja atra* (Chinese cobra), are associated with fatalities primarily due to delayed treatment or secondary infections resulting from severe local tissue damage [34,35]. Effective clinical management of snakebite envenomings in Taiwan necessitates rapid assessment, the prompt administration of species–specific antivenom, and vigilant supportive care to mitigate the potentially fatal consequences of neurotoxic snake venom. Public awareness and education on the recognition of and the role of advanced diagnostic tools in improving management strategies to snakebites are paramount because such responses are essential for reducing both the morbidity and mortality associated with snakebite envenomings in Taiwan.

## Figures and Tables

**Figure 1 toxins-17-00014-f001:**
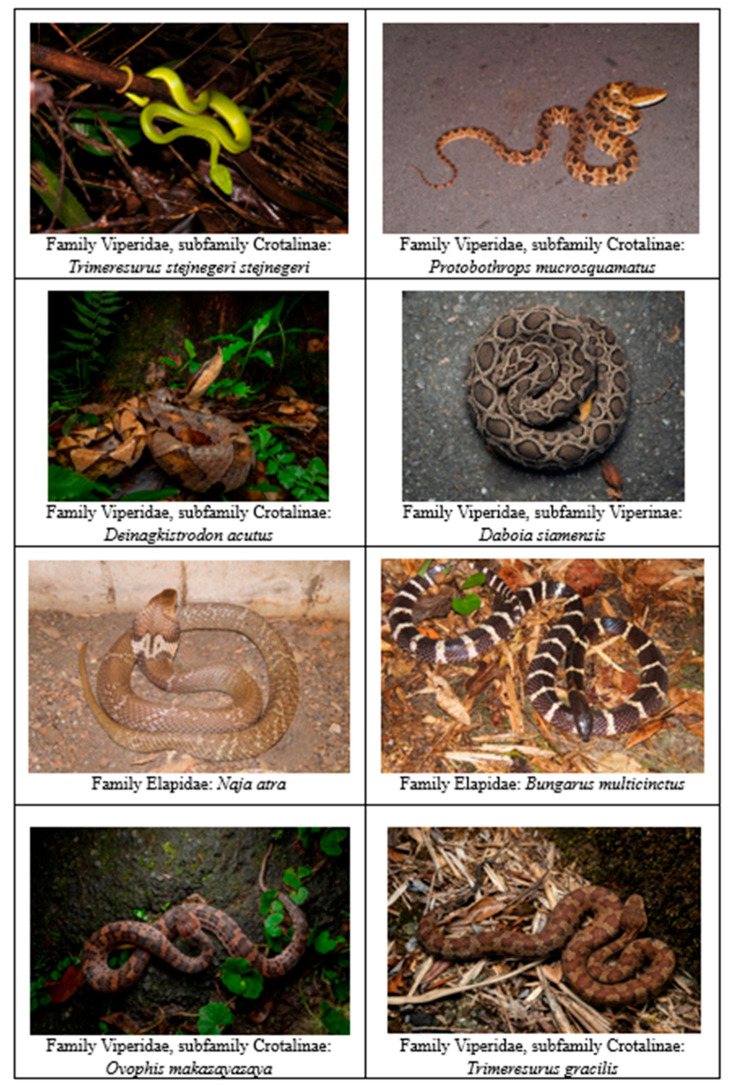
Physical characteristics of venomous snakes in Taiwan. Physical characteristics of eight venomous snake species in Taiwan (*Trimeresurus stejnegeri stejnegeri*, *Protobothrops mucrosquamatus*, *Deinagkistrodon acutus*, *Daboia siamensis*, *Naja atra*, and *Bungarus multicinctus*), along with recently reported crotaline species associated with substantial envenoming in humans.

**Figure 2 toxins-17-00014-f002:**
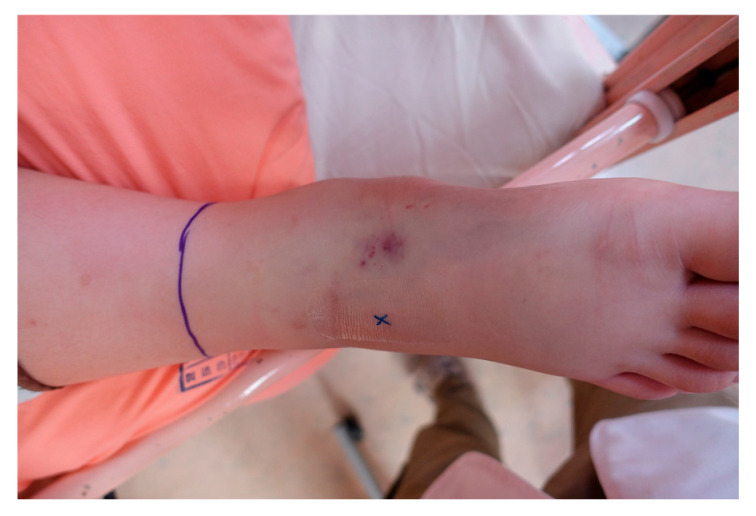
Clinical presentation of *Trimeresurus stejnegeri stejnegeri* envenoming on the right foot, illustrating the rapid onset of local symptoms, including swelling, pain, and ecchymosis at the bite site. These manifestations are the most common effects observed in affected patients.

**Figure 3 toxins-17-00014-f003:**
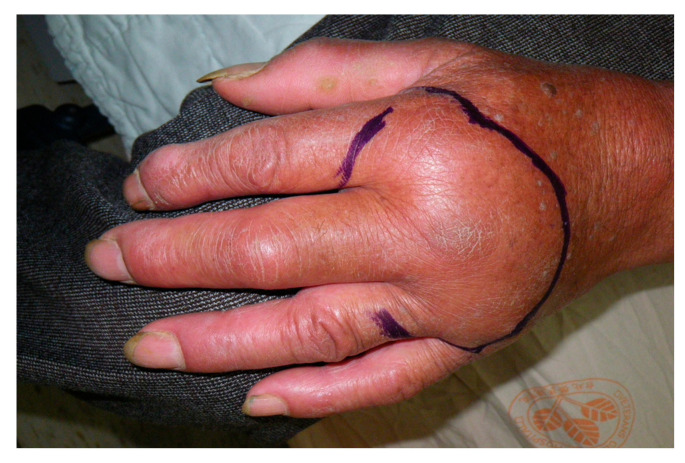
Envenoming by *Protobothrops mucrosquamatus* on the left hand, presenting with tissue swelling and tenderness at the site of envenoming 8 h after the bite. The swelling, caused by the venom’s cytotoxic components, often progresses to involve the entire limb if not promptly treated.

**Figure 4 toxins-17-00014-f004:**
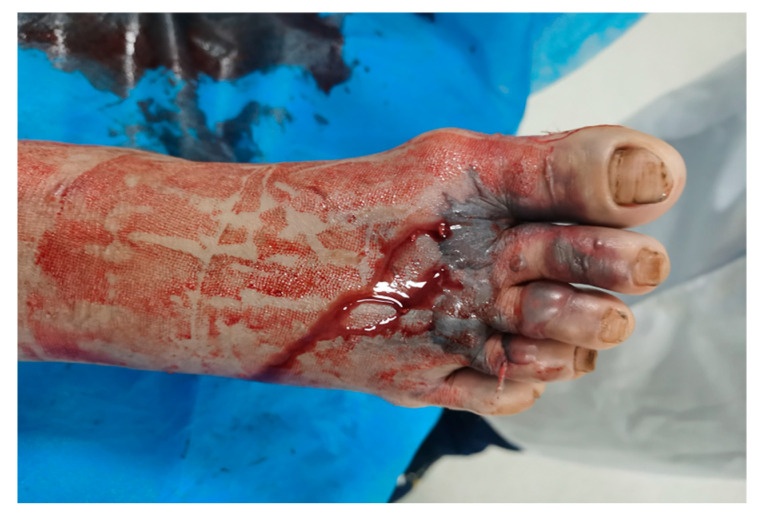
Envenoming by *Deinagkistrodon acutus* on the right foot, with persistent bleeding from fang marks and formation of hemorrhagic bullae 10 h after envenoming. Clinical manifestations typically begin within 2–3 h of the bite and include severe pain, swelling, and the development of hemorrhagic bullae at the bite site.

**Figure 5 toxins-17-00014-f005:**
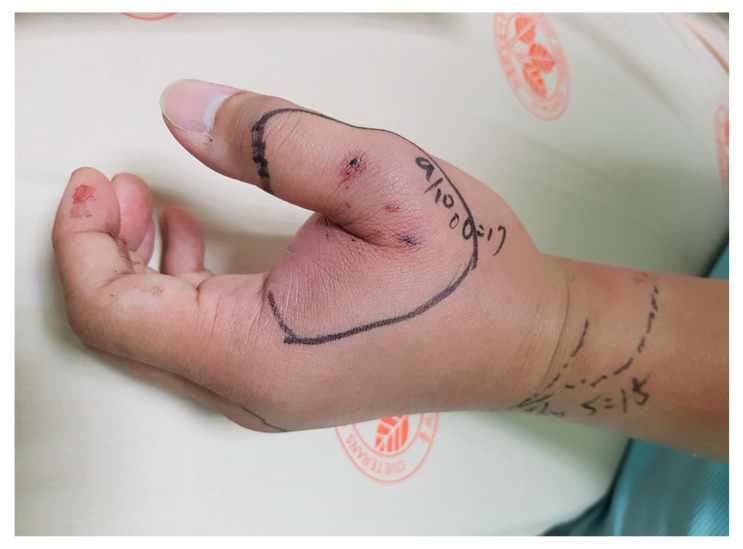
Envenoming by *Daboia siamensis* on the left thumb, displaying localized ecchymosis, is particularly concerning due to the potent hemotoxic effects of its venom, which can trigger coagulopathies and cause immediate and pronounced local symptoms such as swelling, pain, and ecchymosis.

**Figure 6 toxins-17-00014-f006:**
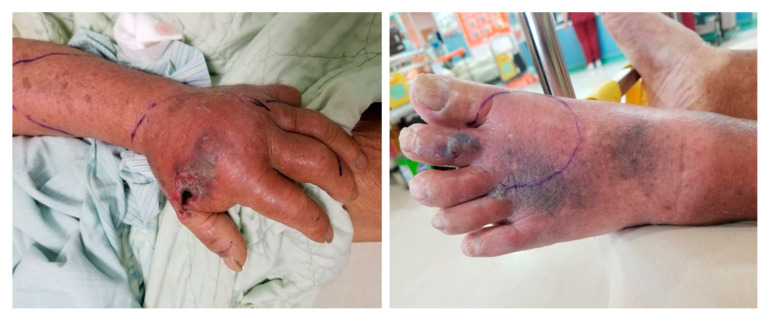
(**Left**) *Naja atra* snakebite on the left hand resulting in local tissue necrosis and bullae formation. (**Right**) *Naja atra* snakebite on the left foot resulting in local tissue necrosis and bullae formation. The clinical manifestations of *N. atra* envenoming are severe local symptoms due to the cytotoxic components of the species’ venom. Patients typically present with localized pain, swelling, and notable tissue necrosis.

**Figure 7 toxins-17-00014-f007:**
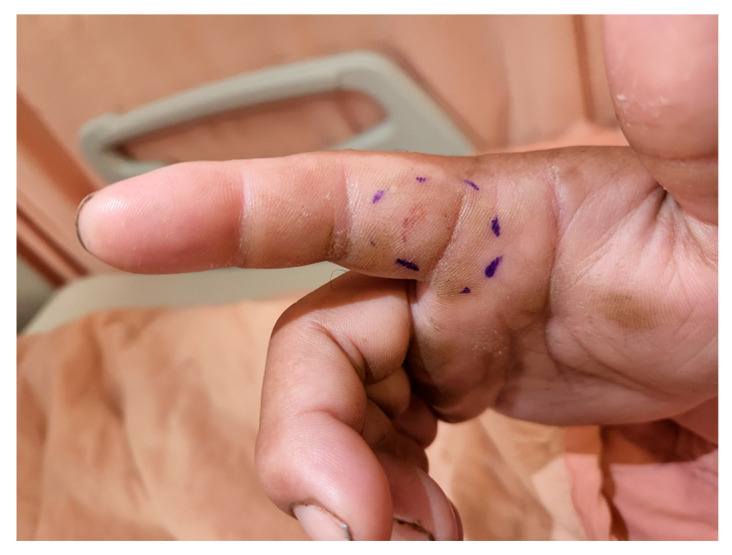
Envenoming by *Bungarus multicinctus* on the right hand, with no noticeable local effects observed. Patients envenomed by *B. multicinctus* typically present with minimal local symptoms at the bite site, such as slight swelling and pain.

## Data Availability

No new data were created or analyzed in this study. Data sharing is not applicable to this article.

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
