# Peer review of "Clinical Characteristics of Snakebite Envenomings in Taiwan"

_toxins, 2024, doi:10.3390/toxins17010014_

Round 1

Reviewer 1 Report

Comments and Suggestions for Authors

Snakebites is an unsolved in many countries, including Taiwan. This review describes the main clinical manifestations associated with snakebites in Taiwan. Although it is a valuable piece for venom toxinologists and clinicians, the authors need to revise many points. 

1. The document needs proofreading. Many words appear together, including the title.

2. The abstract needs to be improved. The objective of this review, its importance and contribution to the literature are unclear. A critical conclusion based on the review must be added.

3. The document contains many word repetitions. Please avoiding use the same words closely.

4. Line 15-18. Not all these points were extensively addressed in the present review. The authors must conclude only based on the addressed points.

5. Please use keywords different from the title.

6. The key contribution must be rewritten. The same sentence was included in the abstract.

7.  Many sentences in the document are missing references. For example, lines 26, 26-29, 29-31, 43-45, 45-48, 85-87 , 93-95and so on.

8. Lines 36-39The figure needs to reflect the content described in the manuscript.  Figure 1 includes more than the information described in this sentence.

9. Lines 38-39. What do the authors mean by biological characteristics? Which biological characteristics have been shown in the figure 1?

10. The figures need a title and a brief description. The title 1 does not reflect its content. What are the different biological characteristics?

11. Please include the number of snakebites in Taiwan in the introduction sections.

12. The comment of the health-care costs of snakebites in Taiwan is really interesting, mainly because it is not available for all affected countries. But authors can also compare the impact on Taiwan with other available data from other countries to have an idea of the meaning of substantial, considerable and exacerbated. These studies (https://doi.org/10.1016/j.toxcx.2021.100074, doi: 10.1371/journal.pntd.0011699 and https://doi.org/10.1371/journal.pntd.0009464.

13. It would be interesting if the authors can provide the distribution of these medically relevant snakes in Taiwan and also the geographical distribution of snakebites in the introduction section.  

14. Have the proteome of the snake venoms from Taiwan medically relevant snakes studied? The introductions section can benefit from this valuable information and also the clinical characteristics can be better explored if correlated with the venom composition.

14. Figure 2. Please provide a brief description in addition to the title.

15. What is the antivenom or antivenoms available in Taiwan? How is snakebite diagnosed in Taiwan? These information is needed as background in the introduction section.

16. What are the studies supporting these sentences (lines 11-114)? They seem more a general view than an evidence-based sentence from Taiwan.  The same is valid for lines 153-154? Is there any study assessing the long-term sequelae?

17. The tiles of subsections must be revised. It is not necessary to repeat the location.

18. Figure 3. Please add a brief description.  The same applies for figures 4 and 5.

19. Have muscle damage and disabilities being reported after Daboia siamensis bite envenoming in Taiwan?

20. Lines 236. What are the most commonly prescribed antibiotics? In terms of multidrug resistance era, the need of using of antibiotics must be debated.

21. What is the mortality rate in Taiwan? What is the snake associated with the majority of the cases?

22. The conclusion must be based on the revised information and not in general knowledge. The authors have not discussed morbidity and mortality in the manuscript. However,  the conclusion includes these points. In this context, the authors have not provided data regarding the antivenom administration to conclude about the timely administration.

23. The introduction section must include the justification and objective of this study? What are the gaps in knowledge? What makes this review important?

24. The authors must describe the criteria used to review the literature and structure this manuscript.

25. The recognition of common clinical manifestations is valuable for the treatment. However, clinicians cannot ignore underreported or uncommon clinical manifestations. It would be great if the authors could add those as well. For example, for Daboia russelii in India, there are many case reports describing rare conditions. How about Daboia siamensis? Similar rare conditions, such as pulmonary embolism, thrombosis,  bilateral adrenal and pituitary haemorrhages,  rectus sheath haematoma have been observed in Taiwan? It would be valuable, reporting these cases and comparing with Russell’s viper venom.

Reviewer 2 Report

Comments and Suggestions for Authors

The article is very interesting to guide you through the signs and symptoms of different types of snakes in Taiwan. 

Some doubts arose when reading the review that could improve the understanding of clinical management, such as:

In the introduction, could you bring up aspects of when the patient does not bring the snake, how is this assessment and decision on antivenin made?

How is access to antivenin in Taiwan?

In the topics divided by the types of the six different snakes involved in the envenomings, there could be standardization of information as not everyone has the same information described, such as:

- venom composition

- Local and systemic signs

- Classification of accidents and respective number of vials 

- Type of antivenin used

- Local and systemic complications of envenoming.

At the end of the article, it could be about the challenges of clinical management of envenoming in Taiwan. If you have other snakes in Taiwan that could interfere with snakebite diagnoses.
